# Longitudinal Association of Maternal Pre-Pregnancy BMI and Third-Trimester Glycemia with Early Life Growth of Offspring: A Prospective Study among GDM-Negative Pregnant Women

**DOI:** 10.3390/nu13113971

**Published:** 2021-11-07

**Authors:** Jiaojiao Zou, Yanting Yang, Qian Wei, Yunhui Zhang, Huijing Shi

**Affiliations:** 1Key Laboratory of Public Health Safety, Ministry of Education, Department of Maternal, Child and Adolescent Health, School of Public Health, Fudan University, Dong’an Road, 130, Shanghai 200032, China; 19111020029@fudan.edu.cn (J.Z.); 18211020111@fudan.edu.cn (Y.Y.); 18111020020@fudan.edu.cn (Q.W.); 2Key Laboratory of Public Health Safety, Ministry of Education, Department of Environmental Health, School of Public Health, Fudan University, Dong’an Road, 130, Shanghai 200032, China; yhzhang@shmu.edu.cn

**Keywords:** pre-pregnancy BMI, third-trimester glycemia, early life growth, GDM-negative pregnant women

## Abstract

Intrauterine modifiable maternal metabolic factors are essential to the early growth of offspring. The study sought to evaluate the associations of pre-pregnancy BMI and third-trimester fasting plasma glucose (FPG) with offspring growth outcomes within 24 months among GDM-negative pregnant women. Four hundred eighty-three mother –offspring dyads were included from the Shanghai Maternal-Child Pairs Cohort. The pregnant women were categorized into four mutually exclusive groups according to pre-pregnancy BMI as normal or overweight/obesity and third-trimester FPG as controlled or not controlled. Offspring growth in early life was indicated by the BAZ (BMI Z-score), catch-up growth, and overweight/obesity. Among those with controlled third-trimester FPG, pre-pregnancy overweight/obesity significantly increased offspring birth weight, BAZ, and risks of overweight/obesity (RR 1.83, 95% CI 1.23 to 2.73) within 24 months. Those who had uncontrolled third-trimester FPG had a reduced risk of offspring overweight/obesity within 24 months by 47%. The combination of pre-pregnancy overweight/obesity and maternal uncontrolled third-trimester FPG increased 5.24-fold risk of offspring catch-up growth within 24 months (*p* < 0.05). Maternal pre-pregnancy overweight/obesity and uncontrolled third-trimester glycemia among GDM-negative women both have adverse effects on offspring growth within 24 months. With the combination of increasing pre-pregnancy BMI and maternal third-trimester FPG, the possibility of offspring catch-up growth increases.

## 1. Introduction

Early life is a sensitive period characterized by adipocyte proliferation; once the number of adipocytes is set, it does not decrease [1]. Globally, the number of overweight/obesity cases is increasing among children under the age of 5 years and even among toddlers, totaling more than 40 million [2], and this weight gain in early life predicts an increasing risk of lifetime chronic morbidity and premature mortality [3,4]. Fetal programming is considered as a key mechanism underlying the association between metabolic programing in utero and negative health outcomes in offspring [5]. The theory of fetal programming emphasizes that pre-pregnancy obesity and poor glycemic control during pregnancy are the main burdens of intergenerational health [6,7]. Intrauterine hyperglycemia exposure and adipogenic intrauterine exposed have the potential for DNA methylation or epigenetic changes in the fetal phenotype; these intrauterine phenomena contributed to the elevated risk of development of glucose intolerance and overweight/obesity in progeny [6,8,9]. The transformation growth factors secreted by fat cells in pregnant women can cause these women to be more prone to blood glucose metabolism disorders [10]. A high maternal pre-pregnancy BMI and abnormal glycemic control are interrelated factors and are associated with the growth of offspring in childhood [11].

The incidence of GDM in China is 17.5% [12], and approximately 55% of women are overweight/obese at the start of their pregnancies [13]. Two recent studies conducted in 1494 Australian and 3805 Dutch participants observed stronger associations between maternal BMI and children growth, and even obesity risk with increasing age [14,15].

Among women with or without GDM, in contrast to a normal pregnancy BMI, pre-pregnancy obesity is a risk factor for overweight/obesity in children [16,17]. Offspring of GDM mothers may experience accelerated fetal growth, and catch-up growth in early life is associated with an increased risk of obesity in school-age children and in adulthood, as several reviews have summarized [18,19]. A meta-analysis of 25 prospective studies [20] showed that higher maternal glucose levels in mid-pregnancy or in late-pregnancy increased the risk of perinatal complications. German researchers found about one-third of obese pregnant women with GDM-negative still has higher glucose level at delivery [21]. Evidence from recent systematic reviews suggested pre-pregnancy overweight/obesity had a greater impact on childhood overweight than GDM [22]. However, the independent and interaction effects of pre-pregnancy BMI and maternal glycemia in late-pregnancy on early-life growth of offspring are not well reflected in the current literature.

Currently, glycemic monitoring is performed mainly in the second trimester, and assessing glycemic control in the third trimester is not yet part of routine medical care for women in most countries [23]; however, the lack of GDM might be regarded as a safe signal and reduce women’s awareness of the risks associated with not controlling weight gain, resulting in poor glycemic control [24]. GDM-negative women are more likely to be neglected by the health system, so more attention should be paid to the health of these mothers and their offspring. Pregnant women with abnormal BMI might develop glucose metabolism disorders in the third trimester. Studies on preterm infants suggest that late pregnancy is a critical period for body fat programming [25] and glycemia imbalance at this time may lead to children being overweight [24]. Maternal FPG, PPG, and non-fasting random samples are appropriate measures of maternal glucose metabolism and are related to adverse birth outcomes [26,27]. FPG reflects the level of basal insulin secretion, furthermore, was an important factor influencing HbA_1c_ levels, which has clinical value [20,28]. More attention to fasting glucose levels may be needed to normalize maternal outcomes [29]. Higher first-trimester fasting glucose levels, even within the nondiabetic range, increase the risk of adverse pregnancy outcomes [20,28]. The impact of intrauterine glycemic control in the third trimester in GDM-negative cases on early growth of the offspring has not been completely determined, and little is known about the effects of both metabolic factors.

We aim to evaluate the associations of pre-pregnancy overweight/obesity and third-trimester glycemia, based on the categorized groups of the two-factor cross combination, in relation to the growth of offspring in early life. In addition, we wanted to assess whether pregnancy management recommendations for healthy overweight/obesity women need to be optimized.

## 2. Materials and Methods

### 2.1. Study Samples

Participants were selected from the ongoing Shanghai Maternal-Child Pairs Cohort (Shanghai MCPC), which began in April 2016 (criteria and protocols for enrollment have been described before [30]. From May 2016 to December 2019, a total of 2306 pregnant women in early pregnancy were included from one regional maternity hospital. No pregestational diabetes mellitus was included as one of the inclusion criteria. We followed offspring from birth to 1, 2, 4, 6, 9, 12, 18, and 24 months postpartum to obtain their information in the process of growing up and Appendix A showed the details of the participants’ selection process (Appendix A). Finally, the current study included 483 mother–child pairs without GDM. A total of 483 term (gestation 37~42 weeks) singletons completed data collection on anthropometric measures. We computed the sample by PASS15.0 software bases on cohort study formula [31], and included participants (*n* = 483) who met the criteria to detect significant statistical difference. No significant difference was found between characteristics of the study participants compared to the subset used for complete analysis in Appendix A. Comparison of the glucose levels of the first trimester and second trimester of the two groups of pregnant women, whether the third trimester blood glucose measurement was performed, no statistical differences was found. Data on sociodemographic and lifestyle factors of mother–child pairs were collected via questionnaire and other medical information from medical record abstraction in the Maternal and Child Healthcare System. Our study protocol was approved by the Institutional Review Board in Public Health School of Fudan University in April 2016 and March 2020, respectively (IRB number 2016-04-0587 and IRB number 2016-04-0587-EX). Written consent was obtained from each patient or subject after full explanation of the purpose and nature of all procedures used.

### 2.2. Glucose Level Assessment

Pregnant women underwent a 75 g oral glucose tolerance test, including measurements of FPG and 1 h and 2 h plasma glucose levels, at approximately 24 weeks gestation. GDM was defined using IADPSG criteria, which are more suitable for the diagnosis of gestational hyperglycemia in China [32]. FPG in the third trimester was the blood glucose level measured by blood taken before breakfast after overnight fasting (at least 8–10 h without any food, except drinking water). It was tested (Hexokinase method [33]) between the 32nd and 33rd gestational weeks, and all glucose level tests were performed by professional medical staff during prenatal examination. According to the “Guidelines for Diagnosis and Treatment of Pregnancy with Diabetes 2014”, the goal of FPG control during pregnancy ≤5.3 mmol/L was defined as third-trimester FPG being controlled, and >5.3 as third-trimester FPG not controlled.

### 2.3. Anthropometric Measures

Infant anthropometric data were body length and body weight of the offspring at birth and at eight subsequent time points (1, 2, 4, 6, 9, 12, 18, and 24 months). The Z score of growth was calculated with the WHO Anthro software (http://www.who.int/childgrowth/software/en/). Birth-weight-standard-deviation scores, standardized as the Z score for sex and gestational age, were calculated. The catch-up growth category in early life was divided into two groups based on the BAZ (BMI Z-score) increment: catch-up growth (≥0.67). A gain in BAZ greater than 0.67 SD scores between the zero and different months indicated clinically significant catch-up growth, and evidence suggested that children with catch-up growth in early life had a higher risk of central obesity at 5 years of age [34]. Overweight/obesity was defined as BAZ > 1, and normal BMI was defined as −2 ≤ BAZ ≤ 1 [35].

### 2.4. Pre-Pregnancy BMI

Data for height and pre-pregnancy weight were obtained from the pregnant women’s baseline interview at 12–16 weeks of gestation, which were highly correlated with the weight and height that the researchers subsequently measured around the 21 weeks of gestation (correlation coefficient = 0.94 for weight and 0.97 for height). Height was measured in m (two decimal places) and weight in kg (one decimal place), was recalled by the pregnant women and surveyed by a nurse. Body mass index (BMI) was calculated by dividing body weight (kg) by height-squared (m^2^), with the results determined with one decimal place. Maternal pre-pregnancy BMI was divided into two groups (normal BMI < 24.0, pre-pregnancy overweight/obesity ≥24.0) by the Chinese BMI cutoffs [36].

### 2.5. Covariates

We retrieved information on maternal characteristics (parity (0, ≥1), age at delivery, last menstrual period, blood pressure taken within 1 month before delivery, prenatal weight) from medical records. In addition, data on educational level, maternal smoke or alcohol during pregnancy, family income, delivery mode, mode of infant feeding within the first 6 months, breast feeding duration, physical activity level, and anxiety and depression in the third trimester were collected by means of professional questionnaires [37,38,39]. Energy intake in late pregnancy was calculated by the food exchange unit using the information collected by semi-quantitative Food Frequency Questionnaire [40]. Gestational age at delivery was calculated as days between the self-reported date of the first day of the last menstrual period and the date of delivery. Gestational weight gain was calculated as the difference between pre-pregnancy and prenatal weight. According to the JNC7 guidelines [41], gestational hypertension was defined as hypertension (≥140/90 mm Hg).

### 2.6. Statistical Analyses

We categorized women into four mutually exclusive groups according to categories of pre-pregnancy BMI and FPG in the third trimester: Group 1 (normal pre-BMI and third-trimester FPG being controlled), Group 2 (overweight/obesity with pre-BMI and third-trimester FPG being controlled), Group 3 (normal pre-BMI and third-trimester FPG not controlled), and Group 4 (overweight/obesity with pre-BMI and third-trimester FPG not controlled). Differences among mean continuous variables by BMI/GDM group were evaluated using ANOVA with Dunnett’s or Bonferroni’s significant difference adjustment for multiple comparisons. Pre-pregnancy BMI and third-trimester glycemia were transformed into continuous variables, and the interaction term of both variates was added to the fully adjusted model to investigate the potential effect by multivariate regression. For all months, overweight/obesity rates of offspring were over 10%. To avoid using odds ratios (ORs) and overstating the risk, we calculated relative risks (RRs) instead by log-binomial regression or Poisson regression with robust variance estimation and linear regression models to evaluate the differences in offspring anthropometric outcomes (BAZ, catch-up growth category, overweight/obesity) among different monthly time intervals between offspring in different groups. Generalized estimating equations were used to study the longitudinal development of offspring. Confounders were chosen based on previous studies and backward selections in primary analyses. Collinearity among the exposure variables was examined, with a maximum variance inflation factor of 2, suggesting no evidence of collinearity. Missing covariate data were imputed using multiple imputation. Significance tests were two-tailed, and a *p*-value < 0.05 was considered statistically significant. All of the analyses were performed with the statistical software packages R 3.4.3 and SPSS (v22.0).

## 3. Results

### 3.1. Participants

Of all studied GDM-negative pregnant women, the mean pre-pregnancy maternal BMI was 21.3, with 18.6% of women being overweight/obesity, and the proportion of women in the third-trimester FPG not-controlled group was 8.1%. Table 1 showed that regardless of the status of third-trimester FPG, pregnant women in the normal pre-BMI group (G1, G3) had a longer gestational weeks than the overweight/obesity group (39.5 ± 1.0 vs. 39.3 ± 1.0, 39.1 ± 1.4 vs. 36.6 ± 4.4), and cesarean section rates were lower. Compared to those in the third-trimester FPG uncontrolled group (G3, G4), those in the third-trimester FPG controlled group had higher birth weights. Regarding to the controlled third-trimester FPG group, the overweight/obesity group had a higher incidence of overweight/obesity offspring at six months of age than the normal group. In the group of third-trimester FPG not controlled, the offspring of pregnant women who were overweight/obesity before pregnancy had a lower BAZ at 9 months than women with normal pre-BMI (data not shown). In a pairwise comparison, between groups with third-trimester FPG control, the pre-pregnancy overweight/obesity group had significantly higher third-trimester FPG than the normal pre-BMI pregnant group.

### 3.2. Effects of Pre-Pregnancy BMI and Third-Trimester FPG on Offspring Growth

To identify the independent and combined effects of pre-pregnancy BMI and third-trimester FPG on offspring anthropometric outcomes within 24 months, Table 2 indicated that pre-pregnancy BMI was positively associated with offspring birth weight and BAZ across the first 24 months of life. For each additional unit in pre-pregnancy BMI, there was an associated increase in birth weight by 100 g, BAZ by 0.34 and increased the risk of overweight/obesity offspring within 24 months by 1.41-fold, while each increment of one unit in the third-trimester FPG decreased the RRs of overweight/obesity offspring by 48% (RR 0.52, 95% CI 0.29 to 0.93). Uncontrolled third-trimester FPG increased the risk of catch-up growth in offspring, although the effect was not significant. The interaction significantly increased the risk of catch-up growth in offspring within 2 years of age (RR 5.24, 95% CI 1.78 to 15.43, *p* = 0.003).

### 3.3. Further Compared the Independent and Combined Effects of the Four Groups of Pre-Pregnancy BMI and Third-Trimester FPG on Offspring Growth

Forest plot of the effect of maternal pre-pregnancy BMI and third-trimester FPG on offspring growth and development from 1–24 months according to groups is shown in Figure 1. After comparing the group of GDM-negative pregnant women with third-trimester third-trimester FPG controlled, we detected pre-pregnancy overweight/obesity gained 90 g of birth weight (*p* < 0.05) and had an increased BAZ in the first 9 and 12 months of life, as well as an increased risk of overweight/obesity in offspring at 9 months of age.

We further compared the independent and combined effects of the four groups of pre-pregnancy BMI and third-trimester FPG on offspring growth (Table 3). If maternal third-trimester third-trimester FPG was controlled, overweight/obesity before pregnancy was significantly associated with a higher BAZ (β = 0.39, 95% CI 0.17 to 0.62) and the risk of overweight/obesity of offspring went up 1.83-fold within 24 months (RR 1.83, 95% CI 1.23 to 2.73) after adjusting for gestational weight gain.

Among a group of pregnant women with normal pre-pregnancy BMI, third-trimester third-trimester FPG not controlled was a protective factor against overweight/obesity offspring within 2 years of age, reducing the risk by 47% (RR 0.53, 95% CI 0.30 to 0.94). However, it seemed to increase the risk of offspring catch-up growth by 1.17 times, though this was not significant. In Model 2, an additional adjustment was made for gestational weight gain, but the results were almost the same as those in Model 1. The combination of pre-pregnancy overweight/obesity and maternal third-trimester FPG not controlled increased the risk of offspring catch-up growth within 24 months after birth (RR 3.49, 95% CI 1.08 to 10.27).

## 4. Discussion

The present study focused on pregnant women without GDM. When women had a controlled FPG status, pre-pregnancy overweight/obesity might still significantly increase birth weight, the BAZ and the risk of the offspring being overweight/obesity during the first two years of life. Uncontrolled third-trimester glycemia can reduce the risk of overweight/obesity among offspring within two years after birth but can increase the likelihood of catch-up growth, even among the offspring of women who had a normal pre-pregnancy BMI. The combination of these two factors has a greater impact on the likelihood of offspring catch-up growth than each factor alone.

The birth weight of newborns increased significantly with increasing pre-pregnancy BMI after controlling the third-trimester FPG, similar to another study [42]. One of the underlying reasons was that the intrauterine high-fat environment caused by pre-pregnancy obesity can stimulate the placenta to secrete abnormal growth factors and lead to excessive growth of the placenta and fetus [43]. However, the explainable mechanisms could not fully explain result among uncontrolled third-trimester FPG women, when the birth weight of newborns decreased with increasing pre-pregnancy BMI. Another possible explanation for this situation was that pre-pregnancy maternal BMI exhibits a much stronger influence than abnormal blood glucose tolerance on offspring growth [44]. Wang et al. found [45] that serum IGF-I levels in offspring of GDM pregnant women were positively correlated with physical development, with the lowest level at 7–8 months after birth and a slow rise at 9 months. The offspring’s BMI peaked at approximately 9 to 12 months of infancy and then declined to its lowest value at the age of 4 to 6 [46]. Different theories exist in the literature regarding whether adjusting for maternal overweight and/or obesity can influence the effects of GDM on offspring with higher BMI in childhood. Maternal pre-pregnancy overweight/obesity [47] may be the strongest predictor of later offspring obesity independent of maternal GDM. Some researchers have taken pregnant women’s pre-pregnancy BMI as an indicator of genetic susceptibility [48], and around 4436 participants in US study showed offspring obesity at ages 6–11 and 12–19 years were associated almost entirely with mothers who were overweight/obesity in early-pregnancy [49].

Systematic reviews [22] have investigated the relationship between the existing maternal hyperglycemia or GDM and the growth trajectory of the offspring was attenuated or disappeared after adjusting for maternal pre-pregnancy BMI. One explanation was that GDM-negative women with high pre-BMI may have offspring with overnutrition. Our study found that the effects showed only a small difference before and after adjustment for gestational weight gain in pregnant women with third-trimester FPG controlled group. This finding is consistent with findings made in The Netherlands, which turned out that within the BMI categories, there was only a small effect of gestational weight gain on offspring overweight/obesity [50].

Monitoring glycemic control in the third trimester plays a relevant role in children’s health [24]. Surveys such as that conducted by Kelly have shown that children whose mothers had diabetes had lower weight at 12 months of age than those without [51]. Other studies suggested [48,51] that the overall BMI growth trajectory showed a downward trend in offspring exposed to GDM subjects from birth through the early 24 months of age, while after 2 years of age, the offspring of GDM mothers still had lower rates of obesity. Together, these studies indicated that the risk of overweight/obesity during early life may be reduced among those whose third-trimester glycemia is not controlled.

Previous research [52] established that offspring exposed to maternal hyperglycemia in utero had early catch-up growth compared to unexposed offspring, which was consistent with our research. Subsequently, offspring showed poor growth or weight loss in the first year after birth, and then, their BMI were higher during adolescence. Follow-up studies in children under 2.5 years of age did not find a significant impact of impaired maternal glucose metabolism in pregnancy on offspring obesity in the Belfast U.K. center of the multinational HAPO Study, and the difference was reflected in school-aged children [53,54]. The underlying explanation could be that after birth, the offspring of women with hyperglycemia during pregnancy are no longer exposed to the drivers of enhanced intrauterine growth, and their growth rate slows down, thus returning to the original genetically programmed growth rate. The intrauterine growth patterns of GDM start accelerating again after more than five years of age [55], coupled with the influence of a poor dietary environment in childhood, leading to an increased risk of overweight. Investigators in Switzerland indicated [56] that offspring growth linked to maternal glucose level, was also affected by other factors such as the placental capacity for glucose transport and placental glucose metabolism. Studies revealed [57,58] that GLUT-1 (glucose transporter-1, GLUT-1) and GLUT-4 expression was decreased owing to hyperglycemia, which led to a limitation of glucose transport across the placenta and a decrease in glucose utilization capacity of the offspring. This condition resulted in an uneven distribution of nutrients between the mother and fetus, which may be associated with reduced overweight/obesity in early life of offspring. The correlation between maternal hyperglycemia (irrespective of the time of onset) and offspring metabolic outcomes is much more complicated and remains unclear, yet, the specific mechanism still requires further illustration.

Summary of previous research indicate that intrauterine hyperglycemia, which includes GDM and controlled third-trimester FPG, is related to childhood overweight/obesity in offspring [7,24] and is associated with metabolic syndrome in adulthood among offspring. The overall BMI growth trajectory of the offspring of GDM pregnant women begins with a downward trend in early life and changes to a gradual increase in childhood, and if the glycemia of the GDM-negative pregnant women was not well controlled in the third trimester, the BMI developmental trajectory of the offspring may be consistent with that of the mother with GDM. Additionally, both the Hyperglycemia and Adverse Pregnancy Outcomes study [59] and the case-control study [60] reported that any degree of elevated blood glucose during pregnancy would increase the risk of long-term insulin resistance and metabolic diseases in offspring, so we speculate that the offspring of pregnant women who have poor blood glucose control in the third trimester may increase the risk of chronic diseases in adulthood, even if they do not have abnormal BMI in early life.

Globally, the highest prevalence of GDM was reported in Europe (31.0%) [61], which meant there are many GDM-negative pregnant women who are easily overlooked by the health system. Glycemic control in late-pregnancy as part of glucose management during pregnancy is unknown, so our study has some clinical value, which is one of our advantages. The validity of the extrapolation of the study needs to be further validated.

The longitudinal study allows for more latitude in discerning causality vs. association. By combining pre-BMI and FPG in the third trimester into groups, we explored the effects of the two factors on the growth of offspring within 24 months. We adjusted for several potential confounders, such as family history of diabetes, gestational weight gain, gestational physical level, gestational hypertension, anxiety and depression in the third trimester, and socioeconomic parameters, to improve the reliability of the results. Second, the subjects of our study were Chinese mother–child pairs, which minimized the confounding effects by ethnic background. There were still some limitations. Some subjects were lost to follow-up at multiple time points in early life, and the relatively small number of pre-pregnancy overweight/obesity women in the third-trimester FPG uncontrolled group may limit our power to detect significant differences among the groups for growth outcome comparisons and lead to unstable estimates. However, the use of longitudinal models would subsequently have resulted in increased power. Due to the limitation of the number of overweight and obese women before pregnancy in this study, they are combined for analysis. Being overweight or having obesity may have a differential impact on offspring, but it still needs to be verified by future studies. If pregnant women received treatment (such as dietary or physical activity guidance) for elevated glucose, associations with outcomes could be impacted by the treatment. Women would not get special treatment (such as dietary and physical activity guidance) if they were not diagnosed as GDM; meanwhile, we adjusted the energy intake and physical activity level in late-pregnancy to maintain the credibility of the results. We regret not having collected information of children’s own energy intake, as it has been recognized as an important factor for children growth and development. Additionally, the effect of blood glucose levels during pregnancy on the development of offspring is also reflected in indicators such as fat percentage and waist circumference, but we did not have data for these indicators. The use of FPG as a study indicator also has some clinical significance; whether FPG or PPG is more sensitive needs to be further studied in the future. Children in the Shanghai MCPC will be followed up until five years of age, and these observations will be used to evaluate the long-term repercussions of our findings.

## 5. Conclusions

Both maternal pre-pregnancy overweight/obesity and uncontrolled third-trimester glycemia have negative effects on offspring growth within 24 months, which were mainly manifested in the increased risk of offspring overweight/obesity and increasing the probability of catch-up growth, respectively. Future primary prevention of childhood overweight/obesity needs to focus on screening for overweight/obesity women planning for pregnancy. At the same time, a negative GDM test during pregnancy is not the only signal of normal blood glucose levels; additional attention may be paid to the third-trimester glycemia.

## Figures and Tables

**Figure 1 nutrients-13-03971-f001:**
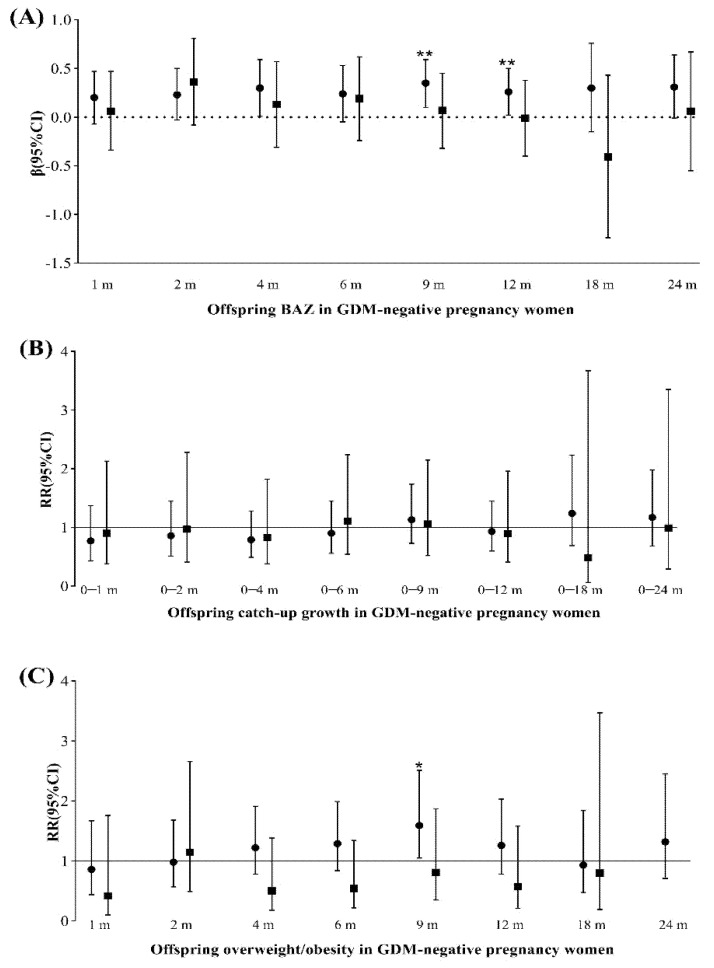
Forest plot of β/RRs with 95% CIs of anthropometric outcomes of 1- to 24-month-old offspring based on groups by maternal pre-pregnancy BMI and third-trimester FPG among GDM-negative women. (**A**) Offspring BAZ in GDM-negative pregnancy women; (**B**) Offspring catch-up growth in GDM-negative pregnancy women; (**C**) Offspring overweight/obesity in GDM-negative pregnancy women. Circles show G2 vs. G1: overweight/obesity with pre-BMI and third-trimester FPG being controlled vs. normal pre-BMI and third-trimester FPG being controlled. Squares show G3 vs. G1: normal pre-BMI and third-trimester FPG not controlled vs. normal pre-BMI and third-trimester FPG being controlled. Because the G4 group had fewer people at different months of age, no analysis was made. ** *p* < 0.01, * *p* < 0.05. Model 1 for offspring birth weight, Model 2 for offspring BAZ, BAZ catch-up growth, Model 3 for offspring overweight/ obese. Model 1: adjusted for: sex, age at delivery, gestational weight gain, education, family income, parity, delivery mode, PA level, energy intake, anxiety and depression in the third trimester, maternal smoke or alcohol during pregnancy, gestational age at delivery Model 2: adjusted for covariates in Model 1 + mode of infant feeding within the first six months (exclusive breastfeeding, mixed feeding, exclusive formula feeding), breast feeding duration. Model 3: adjusted for covariates in Model 2+ Birth weight Z-scores.

**Table 1 nutrients-13-03971-t001:** Characteristics of pregnant-offspring pairs according to four groups by pre-pregnancy BMI and third-trimester FPG (*n* = 483).

Variable	Normal Pre-BMI and Third-Trimester FPG Being Controlled (G1)	Overweight/Obese and Third-Trimester FPG Being Controlled (G2)	Normal Pre-BMI and Third-Trimester FPG Not Controlled (G3)	Overweight/Obese and Third-Trimester FPG Not Controlled (G4)	*p*-Value
N	361	83	32	7	
**Maternal characteristics**					
Age at delivery, mean ± SD, (years)	28.0 ± 4.0	28.5 ± 4.4	27.6 ± 3.5	27.7 ± 1.8	0.691
Gestational weight gain, mean ± SD, (Kg)	14.7 ± 4.8	13.0 ± 5.7	15.3 ± 4.8	12.0 ± 5.7	0.021
FPG in the third trimester, mean ± SD, (mmol/L)	4.0 ± 0.5 ^bcd^	4.1 ± 0.5 ^acd^	6.4 ± 1.3 ^ab^	6.7 ± 1.4 ^ab^	<0.001
Gestational hypertension	45 (12.5)	16 (19.3)	8 (25.0)	4 (57.1)	0.002
Pre-pregnancy BMI, mean ± SD, (kg/m^2^)	20.2 ± 1.9 ^bd^	26.2 ± 2.2 ^ac^	19.9 ± 1.8 ^bd^	26.0 ± 2.4 ^ac^	<0.001
Pre-pregnancy BMI category					<0.001
18.5–23.9	290 (80.3)	0 (0.0)	27 (84.4)	0 (0.0)	
<18.5	71 (19.7)	0 (0.0)	5 (15.6)	0 (0.0)	
≥24.0	0 (0.0)	83 (100.0)	0 (0.0)	7 (100.0)	
Education level > 9 (years)	303 (83.9)	62 (74.7)	30 (93.8)	6 (85.7)	0.074
Family income ≤ ¥200 thousand (RMB)	296 (82.0)	70 (84.3)	26 (81.2)	6 (85.7)	0.953
Parity-nulliparous	218 (60.4)	44 (53.0)	21 (65.6)	4 (57.1)	0.556
PA level in the third trimester as low	223 (61.8)	49 (59.0)	21 (65.6)	4 (57.1)	0.952
Anxiety in the third trimester	44 (12.2)	2 (2.4)	4 (12.5)	0 (0.0)	0.048
Depression in the third trimester	36 (10.0)	6 (7.2)	3 (9.4)	0 (0.0)	0.722
**Offspring** **characteristics**					
Gestational age at delivery, mean ± SD, (weeks)	39.5 ± 1.0	39.3 ± 1.0	39.1 ± 1.4	36.6 ± 4.4	<0.001
Gender-Male	171 (47.4)	49 (59.0)	16 (50.0)	4 (57.1)	0.280
Delivery mode-cesarean section	179 (49.6)	50 (60.2)	4 (12.5)	2 (28.6)	<0.001
Mode of infant feeding within the first 6 months-exclusive breastfeeding	76 (21.1)	21 (25.3)	2 (6.2)	0 (0.0)	0.173
Breast feeding duration, mean ± SD, (months)	4.0 ± 3.7	4.0 ± 3.6	3.5 ± 3.7	3.4 ± 3.6	0.854
Birth weight, mean ± SD, (kg)	3.4 ± 0.4	3.5 ± 0.4	3.3 ± 0.4	2.8 ± 0.9	<0.001

Data are shown as *n* (%) unless otherwise indicated. Based on x^2^ test, with Fisher exact test used for variables with any cell count <10, or Kruskal-Wallis test for continuous variables, *p* < 0.05. ANOVA adjusted with Bonferroni significant difference for multiple comparisons. ^a^ Significantly different than G1. ^b^ Significantly different than G2. ^c^ Significantly different than G3. ^d^ Significantly different than G4.

**Table 2 nutrients-13-03971-t002:** Multivariate regression of offspring anthropometric outcomes from 1 to 24 months according to maternal pre-pregnancy BMI and third-trimester FPG (β/RRs and 95% CI). (*n* = 483).

Variable	Birth Weight(kg)		BAZ		Catch-Up Growth		Overweight/Obesity	
	β (95% CI)	*p*-Value	β (95% CI)	*p*-Value	RR (95% CI)	*p*-Value	RR (95% CI)	*p*-Value
Pre-pregnancy BMI	0.10(0.01,0.19)	0.022	0.34(0.13,0.55)	0.001	1.06(0.66,1.70)	0.812	1.41(1.01,1.97)	0.044
Third-trimester FPG	−0.03(−0.16,0.10)	0.661	0.02(-0.22,0.26)	0.880	1.21(0.64,2.29)	0.563	0.52(0.29,0.93)	0.028
Pre-pregnancy BMI * Third-trimester FPG	−0.18(−0.49,0.13)	0.249	−0.41(−1.19,0.37)	0.300	5.24(1.78,15.43)	0.003	1.00(0.21,4.73)	0.999

Adjusted for age at delivery, gestational age at delivery, parity, breastfeeding duration category, mode of infant feeding with the first six months, PA level, energy intake, anxiety and depression in the third pregnancy, gestational weight gain, gestational hypertension, maternal smoke or alcohol during pregnancy, family history of diabetes, infant sex.

**Table 3 nutrients-13-03971-t003:** The independent and combined effects (β/RRs and 95% CI) of pre-pregnancy BMI and third-trimester FPG on offspring growth from 1 to 24 months in GDM-negative pregnant women by generalized estimation equation. (*n* = 483).

Groups	BAZ		Catch-Up Growth		Overweight/Obesity	
	Model 1	Model 2	Model 1	Model 2	Model 1	Model 2
G1: Normal pre-BMI and third-trimester third-trimester FPG being controlled	Ref.	Ref.	Ref.	Ref.	Ref.	Ref.
G2: Overweight/obesity and third-trimester third-trimester FPG being controlled	0.30 (0.13,0.47)	0.39 (0.17,0.62)	1.04 (0.71,1.52)	0.93 (0.58,1.49)	1.39 (1.01,1.94)	1.83 (1.23,2.73)
G3: Normal pre-BMI and third-trimester third-trimester FPG not controlled	0.07 (−0.16,0.29)	0.03 (−0.22,0.27)	1.25 (0.69,2.29)	1.17 (0.62,2.22)	0.52 (0.30,0.88)	0.53 (0.30,0.94)
G4: Overweight/obesity and third-trimester third-trimester FPG not controlled	−0.01 (−0.63,0.61)	0.05 (−0.67,0.78)	3.22 (0.95,10.31)	3.49 (1.08,10.27)	0.80 (0.19,3.30)	1.04 (0.24,4.52)

Model 1: adjusted for age at delivery, gestational age at delivery, parity, breastfeeding duration category, mode of infant feeding with the first six months, maternal smoke or alcohol during pregnancy, PA level, energy intake, anxiety and depression in the third trimester, gestational hypertension, family history of diabetes, infant sex. Model 2: adjusted for covariates in Model 1 + gestational weight gain.

## Data Availability

Not applicable.

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
