# Peer review of "Longitudinal Association of Maternal Pre-Pregnancy BMI and Third-Trimester Glycemia with Early Life Growth of Offspring: A Prospective Study among GDM-Negative Pregnant Women"

_nutrients, 2021, doi:10.3390/nu13113971_

Round 1
Reviewer 1 Report
Title of Article: Effects of maternal pre-pregnancy BMI and third-trimester glycemia on early life growth of offspring: A prospective study among GDM-negative pregnant women
Hyperglycemia in pregnancy, irrespective of phenotypes, may have adverse effects on both mothers and their offsprings. Fasting blood glucose (FPG) measurement in the first trimester and oral glucose tolerance test (OGTT) in the gestational week 24-28th are among common practices to identify pre-existing overt diabetes (type 1 DM or type 2 DM) and gestational diabetes (GDM), respectively. However, there is still a possibility of overlooking hyperglycemia in the third trimester that could lead to metabolic dysfunction in early life of children. The authors confirmed the rate of uncontrolled FPG in the third trimester, its relationship with adverse outcomes in offspring, and its interaction with maternal pre-pregnancy body mass index (BMI).
1) The novelty of the research
Most of the previous publications investigated the impact of earlier-onset of hyperglycemia, including T1DM, T2DM, and GDM diagnosed from 24-28th gestational week. Only a few papers have assessed the correlation between maternal third-trimester blood glucose only with children outcomes. Even with that, the blood glucose determination of interest was often postprandial glucose rathan than FBG. This current paper evaluated FBG as a valuable and convenient measurement and its interaction with pre-pregnancy BMI in the association with early life growth of offspring.
2) The significance of the results
- Line 262-266: The authors explained the result “The birth weight of newborns increased significantly with increasing pre-pregnancy BMI” by “intrauterine high-fat environment caused by pre-pregnancy obesity can stimulate the placenta to secrete abnormal growth factors and lead to excessive growth of the placenta and fetus”. However, this hypothesis could not fully explain result among uncontrolled 3-trim-FPG women, when the birth weight of newborns decreased with increasing pre-pregnancy BMI.
- Line 279-281: the sentence seemed uncompleted. The authors should explain this point more clearly.
- Line 307-309: the authors explained that “decreased fetal insulin sensitivity” in pregant women with hyperglycemia may be an important reason for reducing overweight and obesity in the early life of offspring. These relationship and explanation should be cautiously interpreted because (1) decreased insulin sensitivy (increased insulin resistance) often leads to weight gain than weight loss because in this condition, hyperinsulinemia leads to increased hunger, increased appettite, higher calorie intake, increased fat storage and decreased fat breakdown; (2) this result was inconsistent with the systematic review and meta-analysis that you referenced [43] and many of the publications, in which exposure to maternal hyperglycemia was associated with offspring obesity. From my point of view, the correlation between maternal hyperglycemia (irrespective of the time of onset) and offspring metabolic outcomes is much more complicated and remains unclear yet.
- In those models (figure 1, table 2, table 3), the authors adjusted for maternal energy intake in late pregnancy. However, children BAZ, catch-up growth, or overweight/obesity rate also depend on their own energy intake, especially after weaning from breastfeeding and replacing breastfeeding with another source of nutrition. This is an important covariate that should be taken into account, or at least be explained in the discussion if the authors could not have the data.
- Although FBG is more convenient to be measured and varies less than postprandial blood glucose (PPG), the correlation between PPG during pregnancy and offspring outcomes is much more apparent than that of FBG. Consequently, controlling blood glucose based on PPG readings had been shown to improve glycemic control better and fewer risks to offspring than using FBG readings (de Veciana, NEJM 1995). Therefore, the inconsistent results coming from FBG in this current study may not reflect the true association and are not convincible enough.
Author Response
Dear reviewer,
Please see the attachment.
Best,
Jiaojiao Zou

Reviewer 2 Report
Manuscript presented by Zou et al. provides some data on the effect of maternal BMI and glycemia on early development of the offspring. The concept is interesting. Nevertheless, the authors did not escape some ambiguities in the text. There are several areas that need to be addressed to improve the manuscript.
Please find below several comments:
The authors should improve the English expressions by replacing them with accurate scientific English. The manuscript needs English proofreading overall, language revisions would be useful in order to make the manuscript more understandable.
Generally, please revise your use if different time forms (present vs. past).
Title: It is worth considering another wording for the title as it is not strictly representative of the study carried out
Introduction:
Line 38-40: The theory of fetal programming, meaning: early programming. Also the term “pre-pregnancy” it is not correct. Please, reconsider using the suitable terms, and go under English proofreading overall.
The introduction can be improved by giving more background on the effect of maternal BMI and adipose tissue structure, as well as maternal and fetal glucose levels and insulin concentrations on the development of the newborns, as it would give a global vision on the importance of the topic on the early programming.
I understand that the study has been performed in in Chinese women, but what is the state of knowledge globally. It appears that authors are expecting data from Chinese women to be vastly different than from any other country. Is such data limited globally, too?
An overview of existing (current) research requires some improvement.
Materials and Methods:
I suggest adding detailed information on the process through a flow chart indicating the stages (e.g. recruitment, follow-up, data analysis), as well as withdrawals along the study, as it would give a better understanding of the process.
I also suggest adding more information about the study design. Ethics Statement (with registration/identification number etc.) in my opinion, should be here.
Line 106: Please add a reference for the Hexokinase method.
Line 130-133: How did you decide on the median BMI as a cut off for high and low BMI?
Statistical section should include description on how the original sample size was calculated in order to detect significant statistical difference.
How did you test for a normal distribution in your variables? And, how did you deal with the repeated measures you have? Please, explain these aspects.
Results:
Please add p-values to indicate significant difference in characteristics of your study participants compared to the subset used for complete analysis.
Tables 1 and 2: Please indicate significant difference in each table
Discussion:
Authors are very focused on comparing their results with other results published in Chinese women. That’s OK, but they could also put the data into a global perspective
The data is more valuable if they could apply it more than just within China. Authors have also included some global data, but adding brief discussion about data available globally, next to the discussion within the country of China, will make this a lot more valuable.
Conclusions:
This section seems more like a missed opportunity. Authors are summarizing their results, which can be included in the conclusions, but they are not providing actual conclusions of their findings. What is the overall message from your study? Or what should be done in future studies?
Author Response

(The authors gave the same response as above.)

Round 2
Reviewer 1 Report
Compared to the first manuscript, the authors have tried to address and revise most of the major concerns raised by the reviewer, and provide additional details: 1) Introduction: - The authors added more evidence showing that the epigenetic changes could explain the adverse effects of intrauterine hyperglycemia and adiposity on fetal phenotypes. - More publications were cited to emphasize the knowledge gap that warrants this current study. - The authors explained more details about the reason for investigating fasting blood glucose instead of other glycemic indexes such as postprandial glucose and non-fasting random glucose. 2) Discussion: - Line 254-257: The authors made an effort to explain the inconsistent results “The birth weight of newborns increased significantly with increasing pre-pregnancy BMI” but “among uncontrolled 3-trim-FPG women, the birth weight of newborns decreased with increasing pre-pregnancy BMI”. The explanation in this current revision is “prepregnancy maternal BMI exhibits a much stronger influence than abnormal blood glucose tolerance on offspring growth” that could be acceptable to a certain degree. - Line 295-301: the reason for reducing overweight and obesity in the early life of offspring in pregnant women with hyperglycemia has been explained more reasonably compared to the first manuscript by the hypothesis of uneven distribution of nutrients between mother and fetus. - Although the authors did not collect children’s energy intake, which is an important covariate that should be considered, they attempted to mention it as a limitation. Minor points: - The terms should be used consistently: “pre-pregnancy” (in the title) versus “prepregnancy” in body text Reviewer Recommendation After addressing and revising major points as suggested, this current revision is appropriate to be published. No significant alterations are needed.
Author Response
Dear reviewer,
Please see the attachment. Thank you very much.
Best,
Jiaojiao Zou

Reviewer 2 Report
The manuscript has been improved significantly. The authors have rearranged and clarified some information.
Authors have given a global perspective to the data by comparing their results to other countries. They have also provided a practical point of view on their data and added what should be done in future studies.
The English language and style has been edited, which makes the manuscript more understandable.
Author Response

(The authors gave the same response as above.)
